# GENERATING HIGH-FIDELITY PRIVACY-CONSCIOUS SYNTHETIC PATIENT DATA FOR CAUSAL EFFECT ESTIMATION WITH MULTIPLE TREATMENTS

## ABSTRACT

In the past decade, there has been exponentially growing interest in the use of observational data collected as a part of routine healthcare practice to determine the effect of a treatment with causal inference models. Validation of these models, however, has been a challenge because the ground truth is unknown: only one treatment-outcome pair for each person can be observed. There have been multiple efforts to fill this void using synthetic data where the ground truth can be generated. However, to date, these datasets have been severely limited in their utility either by being modeled after small non-representative patient populations, being dissimilar to real target populations, or only providing known effects for two cohorts (treated vs control). In this work, we produced a large-scale and realistic synthetic dataset that supports multiple hypertension treatments, by modeling after a nationwide cohort of more than $250,000$ hypertension patients' multi-year history of diagnoses, medications, and laboratory values. We designed a data generation process by combining an adapted ADS-GAN model for fictitious patient information generation and a neural network for treatment outcome generation. Wasserstein distance of $0.35$ demonstrates that our synthetic data follows a nearly identical joint distribution to the patient cohort used to generate the data. Our dataset provides ground truth effects for about 30 hypertension treatments on blood pressure outcomes. Patient privacy was a primary concern for this study; the $\epsilon$-identifiability metric, which estimates the probability of actual patients being identified, is $0.008\%$, ensuring that our synthetic data cannot be used to identify any actual patients. To demonstrate its usage, we tested the bias in causal effect estimation of five well-established models using this dataset. The approach we used can be readily extended to other types of diseases in the clinical domain, and to datasets in other domains as well.

## 1 INTRODUCTION

In health care, studying the causal effect of treatments on patients is critical to advance personalized medicine. Observing an association between a drug (exposure or treatment) and subsequent adverse or beneficial event (outcome) is not enough to claim the treatment is indeed the cause of the observed outcome due to the existence of confounding variables, defined as factors that affect both the treatments and outcomes. Randomized clinical trials (RCTs) have been the gold standard for estimating causal relationships between intervention and outcome. However, RCTs are sometimes not feasible due to logistical, ethical, or financial considerations. Further, randomized experiments may not always be generalizable, due to the restricted population used in the experiments. In the past decade, observational data has become a viable alternative to RCTs to infer causal treatment effects due to both the increasingly available patient data captured in Electronic Health Records (EHRs) (Henry et al. (2016)) and the remarkable advances of machine learning techniques and capabilities. Typically, EHRs capture potential confounding factors such as race, gender, geographic location, eventual proxies of social determinants of health, as well as medical characteristics such as comorbidities and laboratory results.

Many causal inference models have been proposed to estimate treatment effects from observational data. Validation of these models with realistic benchmarks, however, remains a fundamental chal-

lenge due to three reasons. First, the ground truth of treatment effects in a realistic setting is unknown. In real world, we can not compute the treatment effect by directly comparing the potential outcomes of different treatments because of the *fundamental problem of causal inference*: for a given patient and treatment, we can only observe the factual, defined as the patient outcome for the given treatment, but not the counterfactual, defined as the patient outcome if the treatment had been different. Second, legal and ethical issues around un-consented patient data and privacy created a significant barrier in accessing routinely EHRs by the machine learning community. In order to mitigate the legal and ethical risks of sharing sensitive information, de-identification of patient records is a commonly used practice. However, previous work has shown that de-identification is not sufficient for avoiding re-identification through linkage with other identifiable datasets Sweeney (1997); Emam et al. (2011); Malin & Sweeney (2004). Third, most publicly available datasets support either binary or very few treatments, while there has been growing literature developing techniques with multiple treatments in recent years (Lopez & Gutman (2017)).

To address these challenges, in this work we generated a large-scale and realistic patient dataset that mimics real patient data distributions, supports multiple treatments, and provides ground truth for the effects of these treatments. The datasets we generated are synthetic patients with hypertension modeled on a massive nationwide cohort patients' history of diagnoses, medications, and laboratory values. We designed a data generation process by adapting an Anonymization Through Data Synthesis Using Generative Adversarial Networks (ADS-GAN by Yoon et al. (2020))model for fictitious patient information generation and using a neural network for treatment outcome generation. The synthetic dataset demonstrates strong similarity to the original dataset as measured by the Wasserstein distance. In addition, we ensure that the original patients' privacy is preserved so that our dataset can be made available to the research community to evaluate causal inference models.

We demonstrated the use of the synthetic data by applying our dataset to evaluate five models: one inverse probability treatment weighting (IPTW) model, one propensity matching model, one propensity score stratification model all introduced by Rosenbaum & Rubin (1983), and two models in the doubly robust family (Foster & Syrgkanis (2020)).

To our knowledge, this is the first large scale dataset that mimics real data joint distributions with multiple treatments and known causal effects. The approach we used can be readily extended to other types of diseases in the clinical domain, and to datasets in other domains as well.

The rest of this paper is organized as follows. In Section 2, we discuss related works. The details of our method are presented in Section 4. In Section 5, we discuss the evaluation metrics for the quality of the data and results of evaluating several established causal inference models with the data. We discuss the limitations of the work in Section 6 and conclude the paper in Section 7.

## 2 RELATED WORK

Our work is related to several existing works on publicly available databases, fictitious patient record creations, and data generation processes.

### 2.1 PUBLICLY AVAILABLE DATASETS

First used in (Hill (2011)), the Infant Health and Development Program (IHDP) is a randomized controlled study designed to evaluate the effect of home visit from specialist doctors on the cognitive test scores of premature infants. It contains 747 subjects and 25 variables that describe the characteristics of the infants and their mothers. The Jobs dataset by LaLonde (1986) is a benchmark used by the causal inference community, where the treatment is job training and the outcomes are income and employment status after training. The Twins dataset, originally used for evaluating causal inference in (Louizos et al. (2017); Yao et al. (2018)), consists of samples from twin births in the U.S. between the years 1989 and 1991 provided in (Almond et al. (2005)). The Annual Atlantic Causal Inference Conference (ACIC) data challenge provides an opportunity to compare causal inference methodologies across a variety of data generating processes.

Some of the above mentioned datasets do not provide true causal effects. Others are small in size so the models validated on such datasets may perform very differently in a more general real-world setting. All the datasets created for ACIC challenge were designed specifically for competitions. The covariates in the data are either drawn from publicly available databases, or simulated. In the former

case, the datasets are limited by small populations (Du et al. (2021)) and arbitrarily designed data generation processes, which did not aim to capture any real-world causal relationships (Karavani et al. (2019)). In the latter case, the distribution of the data is not realistic, i.e, dissimilar to the distribution of any real dataset.

## 2.2 EHR DATA GENERATION

Walonoski et al. (2018) generated synthetic EHRs based on publicly available information. The focus of their work is on generating the life cycle of a patient and how a disease evolves over time. Goncalves et al. (2020) evaluated three synthetic data generation models–probabilistic models, classification-based imputation models, and generative adversarial neural networks–in generating realistic EHR data. Tucker et al. (2020) used a Bayesian network model to generate synthetic data based on the Clinical Practice Research Datalink (CPRD) in the UK. Benaim et al. (2020) evaluated synthetic data produced from 5 contemporary studies using MDClone. Wang et al. (2021) proposed a framework to generate and evaluate synthetic health care data, and the key requirements of synthetic data for multiple purposes. All of these works focus on EHR data generation producing patient variables but without ground truth for causal effects. In contrast, the focus of our work is not only on generating patient variables, but producing ground truth for causal effects as well.

## 2.3 POTENTIAL OUTCOME GENERATION

To validate their models, many researchers such as (Schuler & Rose (2017)) created synthetic co-variates and produced potential outcomes with a designed data generation process. Such datasets are not designed to approximate any real data distributions. Franklin et al. (2014); Neal et al. (2020) generated potential outcomes from covariates with known causal effects, but without any regard to patient privacy. We addressed the critical issue of patient privacy concerns so our data can be made available for the research community to evaluate their models.

## 3 PATIENT CLAIM DATA AND INCLUSION EXCLUSION CRITERIA

To make our synthetic data realistic, we generate the data based on a real-world patient claim database from a large insurance company in the United States. This database contains 5 billion insurance claims (diagnoses, procedures, and drug prescriptions or refills) and lab test results from 56.4 million patients who subscribed to the company's service within a 5-year time period between December 2014 and December 2020. From this database, we extracted a subset of patients affected by hypertension. We chose hypertension because there are a large number of related claims, making it easier to learn the data distribution. In addition, since it is a condition affecting nearly half of adults in the United States (116 million, or $47\%$), our generated dataset can be directly used for clinical researchers to develop and evaluate their models for this important disease. Patients are included in the dataset if they have a medical claim indicating hypertension (ICD code I10, I11.9, I12.9, and I13.10) or currently treated with anti-hypertensive medications. We exclude patients from the dataset if they are age <18 or age >85, affected by white coat hypertension, secondary hypertension, malignant cancers, dementia, or are pregnant. After applying the above mentioned inclusion and exclusion criteria, we have about 1.6 million patients included in this study. We further exclude patients treated with a combination of drugs rather than a single drug. We then rank the drugs by the number of patients treated with each drug, and only keep patients treated with one of the 28 most popular drugs. These filtering steps produce about $262,000$ patients in the study. The distribution of this dataset is then learned and used to generate synthetic patients, viewed as samples drawn from the learned distribution.

The patients' diagnoses and treatment history and how their conditions evolve over time are captured by trajectory data consisting of labs, diagnoses and their corresponding dates. For the convenience of data processing and analysis, we convert the trajectory data into tabular data with rows representing different patients (samples) and columns representing patient features (variables) including patient demographics, diagnoses, medications and labs. In Table. 1, we list and briefly describe these 60 patient variables: 2 variables (F1) describing the systolic blood pressure before the treatment and the date it was measured, 2 variables (F2) describing the same metric but after the treatment, 3 variables (F3) indicating current and prior drug usage and refill information, 4 variables (F4) describing patient

| Family var. | Var. names | Description |
|---|---|---|
| F1 | date-, lab- | First lab result and date |
| F2 | date+, lab+ | Second lab result and date |
| F3 | drugs, prior_drugs, last_refill | Drugs' info |
| F4 | age, gndr_cd, race_cd, ethncty_cd | Age/Gender/Ethnicity |
| F5 | lab measurement results and date | 11 lab measurements and date |
| F6 | safety_morbs, morbs_prior | Current and previous comorbidities |
| F7 | zip_cd, total_pop, p_female, median_income etc | Zip code and related statistics |
| F8 | trajectory_index, mcid | Meta-information |

Table 1: The patient claim dataset on hypertension contains $1,618,363$ observations and 60 variables. Above the family of such variables are listed and briefly described.

basic information (age, gender, ethnicity), 30 variables (F5) indicating laboratory measurements, 2 variables (F6) indicating the presence or absence of comorbid conditions defined by the Charlson Comorbidity Index (Charlson et al. (1987)), 15 variables (F7) describing the patient's zip code, the racial makeup and income levels in the patient's zip code tabulation area (ZCTA), 2 variables (F8) indicating meta information.

In this study, we are interested in the causal effects of anti-hypertensive drugs (current drugs of F3) on patient outcomes, measured as the difference between the first (F1) and second lab results (F2).

## 4 APPROACH

To generate the synthetic data, we first generate the patient variables using an adapted ADS-GAN model, then generate the treatment outcomes using a neural network. Our approach can be conceptually decomposed into four steps described below.

### 4.1 STEP 1: DATA PREPROCESSING

Since we generate the synthetic data from the patient claim data extracted in Section 3, in this step we preprocess the patient claim data and prepare it for subsequent steps. Described in Table 1, this dataset are of mixed data types: integers (*e.g.*, age), floats (*e.g.*, lab values), categorical values (*e.g.*, drugs), and dates. Further, the values and dates of a lab test are missing for some patients if the lab test was not ordered by the doctors for these patients. In this step, we clean, one-hot encode, transform and standardize the data so that all the variable are transformed into numerical values in $[0, 1]$ range. We also add a binary feature for each lab test to indicate missing lab values and dates. The resulted dataset has 221 features and we call it *the original dataset*, to be distinguished from the synthetic dataset. The details of this step are described in Appendix A.1.

### 4.2 STEP 2: GENERATION OF OBSERVED VARIABLES USING ADS-GAN

In this step we generate synthetic patients characterized by the same variables as listed in Table 1. We want to achieve two goals in this step: to make the synthetic data as realistic as possible and to make sure the probability of identifying any actual patients in the original dataset from the synthetic dataset is very low. We quantitatively define the identifiability in Definition 2, and the realisticity as the Wasserstein distance (Gulrajani et al. (2017)) between the distribution of the synthetic dataset and that of the real dataset it is modeled after.

There is a trade-off between identifiability and realisticity of generated data. Frameworks like MedGan (Choi et al. (2018)) and WGAN-GP (Arjovsky et al. (2017)) do not explicitly define and allow to control the identifiability levels. Therefore, we evaluated generative models that allow to explicitly control such trade-off, *e.g.* the ADS-GAN (Yoon et al. (2020)), PATE-GAN (Jordon et al. (2019)) and DP-GAN (Xie et al. (2018)). ADS-GAN proved to consistently outperform the benchmarks across the entire range of identifiability levels on both the MAGGIC and the three UNOS transplant datasets. It is also based on a measurable definition for identifiability. Another advantage of ADS-GAN is the use of Wasserstein distance to measure the similarity between two high

dimensional joint distributions. We finally selected ADS-GAN and adapted it to generate the patient variables in our study.

Let us denote the original dataset obtained in Section 4.1 as $\mathcal{D} = \{\mathbf{x}_i\}_{i=1}^N$, where $\mathbf{x}_i \in X \subseteq \mathbb{R}^d$ and $\mathbf{x}_i = \left(x_i^{(1)}, x_i^{(2)} \ldots, x_i^{(d)}\right)$, with $x_i^{(j)} \in \mathcal{X}^{(j)} \subseteq \mathbb{R}$ representing the $j$-th feature of a patient. $N$ is the number of samples and $d$ is the number of features of each sample.

Let $P_X$ denote the underlying distribution from which each $\mathbf{x}_i$ is drawn and let $z \sim P_Z$ be drawn from a multi-variate Gaussian distribution. The framework of ADS-GAN is to train a generator $G : X \times Z \to X$ and a discriminator $D : X \to \mathbb{R}$ in an adversarial fashion: the generator $G$ which produces synthetic patient variables $\hat{\mathbf{x}}_i = G(\mathbf{x}, z)$ where $\hat{\mathbf{x}}_i \in \hat{X} \subseteq \mathbb{R}^d$ ensures that the synthetic dataset $\hat{\mathcal{D}} = \{\hat{\mathbf{x}}_i\}$ are not too close to $\mathcal{D}$ as measured by the $\epsilon$-identifiability defined below; on the other hand, the discriminator $D$ which measures the distance between two distributions ensures that the distribution of generated variables $P_{\hat{X}}$ is indistinguishable from the distribution of real variables $P_X$.

**Definition 1.** We define the weighted Euclidean distance $U(\mathbf{x}_i, \mathbf{x}_j)$ between $\mathbf{x}_i$ and $\mathbf{x}_j$ as

$$U(\mathbf{x}_i, \mathbf{x}_j) = \|\mathbf{w}(\mathbf{x}_i - \mathbf{x_j})\|,$$

where $\mathbf{w}$ is a weight vector. Then $r_i$ is defined as

$$r_i = \min_{\mathbf{x_j} \in D/\mathbf{x_i}} U(\mathbf{x}_i, \mathbf{x}_j),$$

where $\mathcal{D}/\mathbf{x_j}$ represents the dataset $\mathcal{D}$ without $\mathbf{x_i}$. From the definition, $r_i$ is the weighted minimum distance between $\mathbf{x_i}$ and any other observations in $\mathcal{D}$.

We similarly define $\hat{r}_i$ as

$$\hat{r}_i = \min_{\hat{\mathbf{x}}_j \in \hat{\mathcal{D}}} U(\mathbf{x}_i, \hat{\mathbf{x}}_j),$$

which is the weighted minimum distance between $\mathbf{x_i}$ and the synthetic samples in $\hat{\mathcal{D}}$.

**Definition 2.** The $\epsilon$-identifiability of dataset $D$ from $\hat{D}$ is defined as

$$\epsilon = \mathcal{I}\left(\mathcal{D}, \hat{\mathcal{D}}\right) = \frac{1}{N} \sum_i \left[\mathbb{1}_{(r_i > \hat{r}_i)}\right]$$

where $\mathbb{1}$ is an indicator function.

To calculate the weight vector $\mathbf{w}$, we first calculate the discrete entropy of the $j$-th feature, *i.e.*

$$H\left(X^{(j)}\right) = - \sum_{x^{(j)} \in \mathcal{X}^{(j)}} P\left(X^{(j)} = x^{(j)}\right) \log\left[P\left(X^{(j)} = x^{(j)}\right)\right]$$

The weight for a feature is then calculated as the inverse of $H\left(X^{(j)}\right)$. In reality, if a patient can be re-identified, the re-identification is more likely through rare characteristics or medical conditions of a patient. Calculating the weight this way ensures that the rare features of a patient are given more weight, correctly reflecting the risk of re-identification associated with different features.

We base the discriminator $D$ on Wasserstein GAN with gradient penalty (Gulrajani et al. (2017)) (WGAN-GP), which adopts Wasserstein distance between $P_{\hat{X}}$ and $P_X$, and defines the loss $\mathcal{L}_\mathcal{D}$ for the discriminator $D$ as

$$\mathcal{L}_\mathcal{D} = \mathbb{E}_{\mathbf{x} \sim P_X, \hat{\mathbf{x}} \sim P_{\hat{X}}} \left[D(\mathbf{x}) - D(\hat{\mathbf{x}}) - \mu\left(\|\nabla_{\tilde{\mathbf{x}}} D(\tilde{\mathbf{x}})\|_2 - 1\right)^2\right] \tag{1}$$

where $\tilde{\mathbf{x}}$ belongs to a random interpolation distribution between $P_X$ and $P_{\hat{X}}$ and $\mu$ is a further hyperparameter that we set to 10 based on previous work (Gulrajani et al. (2017)). We implement both the generator and the discriminator using multi-layer perceptrons.

To train the generator $G$, we need to compute the $\epsilon$-identifiability by computing $r_i$ and $\hat{r}_i$ for every sample, which is computationally expensive. To solve the problem, Yoon et al. (2020) made a simplifying assumption that $G(\mathbf{x}, z)$ is the closest data point to $\mathbf{x}$. However, this assumption can be violated during the training of the network that maximizes the distance between $G(\mathbf{x}, z)$ and $\mathbf{x}$.

We here introduce a contrastive loss (triplet ranking loss, Schroff et al. (2015)) item, which is defined as

$$U_{con}\left(\mathbf{x}, \mathbf{x}', z\right) = \max\left(0, U\left(\mathbf{x}, G\left(\mathbf{x}, z\right)\right) - U\left(\mathbf{x}', G\left(\mathbf{x}, z\right)\right)\right). \quad (2)$$

Then, the final identifiability loss function $\mathcal{L}_{\mathcal{I}}$ is

$$\mathcal{L}_{\mathcal{I}} = \mathbb{E}_{\mathbf{x} \sim P_X, z \sim P_z}\left[-U\left(\mathbf{x}, G\left(\mathbf{x}, z\right)\right)\right] + \beta \mathbb{E}_{\mathbf{x}, \mathbf{x}' \sim P_X}\left[U_{con}\left(\mathbf{x}, \mathbf{x}', z\right)\right]. \quad (3)$$

Similar to (Yoon et al. (2020)), this loss also assumes that $G\left(\mathbf{x}, z\right)$ is the closest data point to $\mathbf{x}$. However, a penalty on the loss function will be imposed if this assumption is violated when the generated sample $G\left(\mathbf{x}, z\right)$ is closer to $\mathbf{x}'$, a randomly drawn sample from dataset $\mathcal{D}$, than to $\mathbf{x}$. The strength of the penalty term is controlled by $\beta$.

In the final optimization problem, we minimize $G$ and maximize $D$ simultaneously, written as

$$G^*, D^* = \arg\min_G \max_D \left[\mathcal{L}_{\mathcal{D}} + \lambda L_{\mathcal{I}}\right] \quad (4)$$

where $\lambda$ is a hyper-parameter that controls the trade-off between the two objectives. Once trained, the adapted ADS-GAN model can be used to produce synthetic data set $\hat{\mathcal{D}}$.

### 4.3 STEP 3: DATA GENERATION MODEL AND CAPTURED CAUSAL EFFECTS

We need to determine a data generation model and proper treatment effects to use in the data generation process to produce the potential outcomes of the synthetic data, i.e, the factuals and counterfactuals. Many researchers use arbitrary functions and arbitrary treatment effects. For example, Schuler & Rose (2017) used a linear function as the data generation process and set the treatment effects arbitrarily. To improve upon such an approach, in this work we train a neural network model from the original dataset aiming to capture the mapping from patient covariates to outcomes as well as the treatment effects. The captured treatment effects are not necessarily the true effects, but serve as the ground truth in the synthetic data when the data is used to evaluate causal inference models because the patient outcomes in the synthetic data are generated from these causal effects.

We first partition the domain of observed variables $X \subseteq \mathbb{R}^d$ into the covariate domain $X_C \subseteq \mathbb{R}^{d_c}$, the treatment domain $X_T \subseteq \mathbb{R}^{d_t}$ and the outcome domain $X_o \subseteq \mathbb{R}$, so that $d \geq d_c + d_t + 1$. The covariates are all the patient variables excluding drugs, prior drugs, zip code, and lab+. Treatments are the drugs. Outcome is the difference between lab+ and lab-. In this step, each treatment $t_i \in X_C$ is one-hot encoded and represented by a $d_t$ dimensional vector, where $d_t$ is the number of treatments. Given a collection of $N$ observations, we use $Y_i\left(t_i\right) \in \mathbb{R}$ to denote the potential outcome of the $i$-th individual, if treated with the treatment $t_i \in X_C$. We assume that $\left(Y_i, t_i, \mathbf{x}_i\right) \in \mathbb{R} \times X_C \times X_C$ are independent and identically distributed, and that the three fundamental assumptions for causal inference (Rosenbaum & Rubin (1983)) in Appendix A.2 are satisfied.

Following (Lopez & Gutman (2017); Shalit et al. (2017)), given $\mathbf{x} \in X_C$ and $t_i, t_0 \in X_C$, where $t_0$ is the placebo, the individual-level treatment effect (ITE) of $t_i$ can be defined as

$$\tau_{t_i}\left(\mathbf{x}\right) := \mathbb{E}\left[Y\left(t_i\right) - Y\left(t_0\right) | \mathbf{x}\right]. \quad (5)$$

Hence, the population average treatment effect for treatment $t_i$ can be defined as

$$ATE_{t_i} = \mathbb{E}\left[Y\left(t_i\right) - Y\left(t_0\right)\right] = \int_{X_C} \tau_{t_i}\left(\mathbf{x}\right) p\left(\mathbf{x}\right) d\mathbf{x}. \quad (6)$$

So the data generation process can be modeled as $Y = \Omega\left(\mathbf{x}, t\right)$, where $\Omega : X_C \times \mathcal{T} \to \mathbb{R}$. The true form of $\Omega$ is unknown and can be complicated. Here we make a simplifying assumption that the representation learned from the covariate domain is separated from the representation learned from the treatment domain. Specifically, let $\Phi : X_C \to \mathcal{R}$ be a representation function and $\mathcal{R}$ be the representation space. We define $Q : \mathcal{R} \times \mathcal{T} \to \mathbb{R}$ so that $\Omega\left(\mathbf{x}, t_i\right) = Q\left(\Phi\left(\mathbf{x}\right), t_i\right)$. Also, we denote with $\tau_{t_i}^Q\left(\mathbf{x}\right)$ the treatment effect estimate of $Q$.

With simplified $\Omega$, we propose a neural network architecture shown in Figure 1 that is able to capture $\Omega$, $\Phi$, and at the same time, calculate the treatment effects. For the covariate domain $X_C$, the network is a fully connected feed-forward neural network with *Relu* as the activation function for all the neurons. For the treatment domain $X_T$, the inputs are encoded treatments directly connected to a neuron with a linear activation. The loss function is the standard mean square error (MSE). We apply a dropout to all the layers and apply a gentle L2 regularization to all the weights of the neural network.

The model $\Omega$ is trained on the original dataset described in section 3, where we have one factual for each observation. Due to the separation of the covariate domain and treatment domain, and with the particular architecture of the ANN shown in Figure 1, the neural network weights for treatment connections can be interpreted as the causal treatment effects. Since there is no interaction between the covariates and treatments, the individual treatment effects and population average treatment effects are the same. Indeed, suppose $w_i$ is the weight for treatment $t_i$, one can show that $w_i$ is indeed the $\tau_{t_i}$ in Equation 5 and the $ATE_{t_i}$ in Equation 6.

Figure 1: Neural network architecture for patient outcome generation and causal effect calculation.

### 4.4 STEP 4: GENERATION OF FACTUALS AND COUNTERFACTUALS ON SYNTHETIC DATA

The domain of variables and all its partitions are the same for the real dataset $\mathcal{D}$ and the synthetic dataset $\hat{\mathcal{D}} = \{\hat{\mathbf{x}}_i : \hat{\mathbf{x}}_i = G(\mathbf{x}_i, z), \mathbf{x}_i \in \mathcal{D}, z \sim P_Z\}_{i=1}^{N}$. Hence, the neural network trained on the original dataset in Step 4.3 can be fed with the synthetic patient variables generated in Step 4.2. The neural network outputs are served as the treatment outcomes for the synthetic data.

Once trained, this neural network is capable of generating all factual and counterfactual treatment outcomes for the synthetic data. For any synthetic patient with covariate $\hat{\mathbf{x}}_j \in X_c$, the potential outcome of any treatment $t_i \in X_C$ can be generated as $\hat{Y}_j(t_i) = \Omega(\hat{\mathbf{x}}_j, t_i) = \Omega(\Phi(\hat{\mathbf{x}}_j), t_i)$. However, instead of generating the potential outcomes of all possible treatments in $X_C$, in this work we only generate two potential outcomes for each patient: the factual outcome corresponding to the treatment produced by the ADS-GAN model, and the counterfactual outcome if the patient had not received any treatment. The reason that we do not want to produce all the counterfactual outcomes is that we want to limit the treatment for each patient only to the one generated by the ADS-GAN model, in order to preserve the treatment assignment mechanism learned from the original dataset.

There is a distinction between the assumptions made in Section 4.3 in determining the treatment effects and the assumptions that our synthetic dataset actually satisfies. Specifically, our synthetic dataset satisfies the SUTVA and unconfoundedness assumption, as we did not model the interactions between patients and we provided all the patient variables in the dataset used to generate the outcomes. Whether it satisfies the positivity assumption, however, depends on the original dataset because the patient assignment mechanism of the synthetic data is learned from the original dataset.

## 5 RESULTS

In this section we evaluate the quality of our synthetic dataset. We show that there is strong similarity in both marginal and joint data distributions between the original and synthetic dataset, and that patient privacy is preserved. We evaluate several causal inference models using our dataset to demonstrate the usage of it.

### 5.1 ANALYSIS OF SYNTHETIC AND ORIGINAL DATA DISTRIBUTIONS

We first show how well the generated synthetic data preserves the joint distribution of the original data. We trained the adapted ADS-GAN model for 10,000 epochs and the outcome neural network for 20,000 epochs to generate the synthetic dataset. We calculated the Wasserstein distance (Villani (2008)) between the joint distribution of this synthetic data and that of the original data to be 0.35. To put this value in the correct perspective, we measured the Wasserstein distance between the original dataset and a randomly generated dataset of the same dimensions. This serves as the baseline scenario. In addition, we randomly split the original dataset into two datasets and measured the Wasserstein distance between them, which is essentially the Wasserstein distance between the dataset and itself and serves as the best case scenario. Our results show that the Wasserstein distance between the synthetic and original data distribution (0.35) is almost as good as that in the best case scenario (0.17), and far more lower than the value in the baseline scenario (8.6).

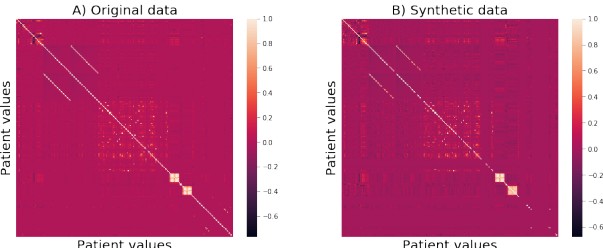

Figure 2: Heatmaps of correlation matrices of patient variables for the original (left) and synthetic data (right), respectively.

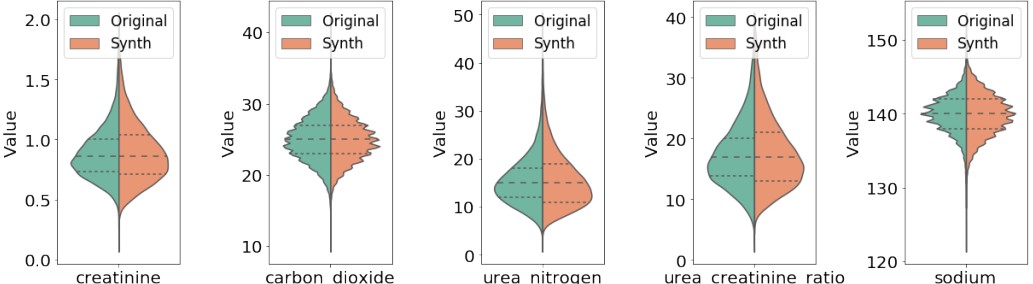

Figure 3: Comparison of marginal distribution of lab values between original and synthetic data. The three horizontal dotted lines in each violin plot from top to the bottom represent the third quartile, median, and the first quartile respectively.

We then compare the joint distributions visually. In Fig. 2, the correlation among all patient attributes in the original (synthetic) dataset is visualized by the heatmap on the left (right). In the heatmap, the brighter the color of a pixel is, the more correlated the two variables are with each other. The diagonal is the brightest in the map, as each pixel on the diagonal represents the correlation between a variable and itself. The two heatmaps show almost identical patterns, indicating the joint distribution of the original data is well preserved in the synthetic dataset.

In Fig. 3, we compare qualitatively the marginal distributions of individual variables of the generated synthetic data (orange) with the related ones from the original data (blue). The figure shows strong similarity between the original and synthetic dataset in both basic statistical summaries (*e.g.*, median and quartiles) and overall shape of these distributions.

## 5.2 IDENTIFIABILITY OF SYNTHETIC DATA

Since the synthetic dataset we generated in this study is meant to be made public, patient privacy has to be preserved to ensure that no actual patients in the original dataset source can be identified through the synthetic dataset. All the synthetic samples in our dataset are conceptually drawn from a distribution, so no single piece of information about any actual patients is directly carried over to our dataset. We further calculate the $\epsilon$-identifiability as defined in Definition 2 to be 0.008%, indicating that the risk of any actual patient being identified is statistically zero.

## 5.3 RESULTS OF TESTING CAUSAL INFERENCE ALGORITHMS ON THE DATASET.

In this section, we evaluate the bias in causal treatment effect estimate of five well established models on our synthetic dataset: two models in the doubly robust (DR) family, one propensity score stratification, one propensity matching, and one inverse probability treatment weighting (IPTW) model. Doubly robust approaches adopt an outcome regression model to estimate the treatment outcome and a propensity model to estimate the probability of a patient being assigned with a treatment. In both the DR models we tested, random forest is used as the outcome regression model. The difference between the two DR models is in the propensity model, which can be logistic regression (DR-LR) or random forest (DR-RF) classification. We use Microsoft DoWhy (Sharma & Kiciman

(2020)) and EconML (Battocchi et al. (2019)) causal inference packages for the implementation. When calculating the causal effect of a treatment, we remove the counter-factuals of this treatment from the dataset to prevent the problem becoming trivial.

We adopted four metrics to evaluate the models: the Spearman's rank correlation coefficient to measure how well the models preserve the rank of the drugs by their treatment effects, Kendall rank coefficient similar to Spearman's coefficient but based on concordant and discordant pairs, correlation between the estimated effects and the ground truth, and finally the $R^2$ to measure how much variance of the ground truth can be explained by the estimate.

To estimate how models perform in a real-world setting, we generated an additional dataset consisting of all patient variables of the original dataset and patient outcomes generated from the trained outcome neural network with patient variables and treatments of the original dataset as its inputs. We call this dataset the hybrid dataset because part of the data comes from the original dataset and part of the data is generated. We run the five causal inference models on the two datasets and re reported all the results in Appendix A.3.

The results on the hybrid dataset show that the algorithms can be grouped into two categories in terms of their performance: the propensity stratification, propensity matching, and two doubly robust models are in the first group and the IPTW model is in the second group. The models in the first group produced much better results than the IPTW model. The results on the synthetic dataset show a similar pattern. The variance of the performance metrics of the models in the first group is larger than that on the hybrid dataset, but all these models perform much better than the IPTW model. Investigating why some models perform better than others on the two datasets is out of scope of this work. Here we show that the synthetic data preserves the relative performance of different models that would be achieved in a more realistic setting, simulated by the hybrid dataset.

## 6 DISCUSSION

In general, inclusion and exclusion criteria applied to the data may introduce selection bias. Our work was designed with a target trial in mind in which patients are recruited at an initial qualifying measurement and then followed up after treatment assignments. We believe this minimizes the impact of selection bias from conditioning on the inclusion and exclusion criteria in our original data. In this work, we only produced one dataset for hypertension and evaluated five causal inference models. We leave it to future work to produce synthetic datasets for other diseases and evaluate and compare other causal inference models. Because hypertension affects almost half of adults in the United States, a synthetic dataset on hypertension is of significant value by itself. For simplicity, in this study we did not consider treatment modifiers, i.e, interactions between treatments and patient variables. It is an interesting and important topic which we plan to address in the future.

## 7 CONCLUSION

To validate machine learning models, researchers have traditionally relied on labeled data, *i.e.* ground truth. Due to the fundamental problem of causal inference, however, the lack of realistic clinical data with ground truth makes it difficult to evaluate causal inference models. In this work, we produced a large-scale and realistic synthetic data by adapting an ADS-GAN model to generate patient variables and using a neural network to produce patient outcomes. The data we generated supports multiple treatments with known treatment effects. We demonstrated that this synthetic dataset preserves patient privacy and has strong similarity to the original dataset it is modeled after. We believe that it will facilitate the evaluation, understanding and improvement of causal inference models, especially with respect to how they perform in real-world scenarios.

### REPRODUCIBILITY STATEMENT

To contribute to reproducibility, we provide in the supplementary material the code for replicating experiments. In section 3 we provide detailed description of the original real-world dataset and related inclusion and exclusion criteria. In section A.1 we provide a detailed description of the preprocessing procedure to make the generation process transparent and reproducible. Finally, we report the the link to download our synthetic dataset in the final paper.

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

# A   APPENDIX

## A.1   STEP 1: DETAILED DESCRIPTION OF DATA PREPROCESSING

Here we provide a description of the preprocessing procedure to clean and transform the patient claim data described in Table 1. The goal of this step is to prepare the data ready for subsequent steps.

We first one-hot encode all the categorical variables in the table, such as gender and race. For each lab test there are two variables: lab values and lab dates. Since all the lab dates were after the 'date-' date, which was the date when the patient's systolic blood pressure was measured before treatment, we convert all the lab dates to the number of days since 'date-', excluding holidays and weekends. The reason for excluding these dates in the convert ion is that we eventually reverse the exact process to convert the synthetic lab dates in number of days back to the original date format, we make sure that such synthetic dates do not fall on holidays and weekends. Note that the exclusion of holidays and weekends does not apply to the prescription dates, assuming prescriptions can be filled on any day. When we transform the 'date-' variable itself, we arbitrarily pick a date proceeding all the 'date-' values, and convert the 'date-' values to the number of days since this picked date, excluding all holidays and weekends.

There are many missing lab dates and values, as expected. If doctors order a certain lab test for a small portion of patients, the lab test and dates for the majority of the patients would be missing. We fill the missing values with the mean of the values that are present. However, this induces a data distribution that significantly deviates from Gaussian distribution: the probability of the mean is much higher than the probability of other values. Our experiments showed that learning such a distribution was difficult. To address this problem, we created a binary indicator for each lab test, which is set to 1 if the lab test is not missing, and 0 otherwise. The intuition is that it is easier for the model to learn not only the distribution of the indicator, but also the join distribution of the indicator and its corresponding lab results, thus enhancing the synthetic data's ability to more faithfully reflect the original data distribution. Another benefit of doing this is to directly preserve the information on whether a lab test was ordered for each patient. Our experiments show that this approach works well in practice. The final step of data preprocessing is to standardize all the values to the $[0, 1]$ range.

## A.2   FUNDAMENTAL ASSUMPTIONS FOR TREATMENT EFFECT ESTIMATIONS

We use $t \in \{t_1, t_2, \ldots\} = X_C$ to denote treatments, $Y_i(t_i) \in \mathbb{R}$ to denote the potential outcome if the patient is treated with the treatment $t_i \in X_C$, and $\mathbf{x}_i$ to denote the variables for patient $i$.

**SUTVA (Stable Unit Treatment Value Assumption)** *Potential outcomes for one individual are unaffected by the treatment of others.*

**Unconfoundedness** *The treatment distribution is conditional independent of the potential outcomes given covariates,* i.e. $(\forall i) \left( t_i \perp\!\!\!\perp \left( Y_i(t_1), Y_i(t_2), \ldots, Y_i\left(t_{|X_C|}\right) \right) \right) | \mathbf{x}_i$.

**Positivity** *Every individual has a non-zero probability to receive each one of the treatments, for a given level of covariates,* i.e. $(\forall i)(\forall s)\left(s \in \mathcal{T} \wedge 0 < P(t_i = s | \mathbf{x}_i) < 1\right)$.

## A.3   RESULTS OF EVALUATING DIFFERENT CAUSAL INFERENCE MODELS

|  | spearmanr | kendalltau | correlation | R2 score |
|---|---|---|---|---|
| Propensity Stratification | 0.99 | 0.92 | 0.99 | 0.99 |
| Propensity Matching | 0.92 | 0.8 | 0.98 | 0.94 |
| Doubly Robust - RF | 0.95 | 0.93 | 0.99 | 0.95 |
| Doubly Robust - LR | 0.99 | 0.96 | 0.99 | 0.99 |
| IPTW | 0.59 | 0.48 | -0.46 | -5.9 |

Table 2: Model evaluation on hybrid data. The results show that the first four models perform much better than the IPTW model.

|  | spearmanr | kendalltau | correlation | R2 score |
|---|---|---|---|---|
| Propensity Stratification | 0.81 | 0.68 | 0.87 | 0.6 |
| Propensity Matching | 0.52 | 0.37 | 0.47 | -11 |
| Doubly Robust - RF | 0.92 | 0.78 | 0.97 | 0.94 |
| Doubly Robust - LR | 0.74 | 0.63 | 0.88 | 0.75 |
| IPTW | 0.09 | 0.04 | 0.06 | -34 |

Table 3: Model evaluation on synthetic data. The results show a pattern similar to the table for the hybrid dataset: the first four models perform much better than the IPTW model. The performance variance of the first four models is larger than that on the hybrid dataset.

