# OpenReview forum: "Generating High-Fidelity Privacy-Conscious Synthetic Patient Data for Causal Effect Estimation with Multiple Treatments"
_ICLR.cc/2022/Conference — ICLR 2022 Submitted_

### Official Review · Reviewer_eqMK · 2021-11-01

**Correctness:** 3
**Technical Novelty And Significance:** 2
**Empirical Novelty And Significance:** 2
**Recommendation:** 5
**Confidence:** 4

**Main Review:**

Strenth:
+ The paper is easy to follow and codes are provided for review.

Weaknesses:
- The proposed problem is not well addressed in this paper. The purpose to simulate patient data is to evaluate causal inference models with generated ground truth. However, it is hard to guarantee that the generated treatment effects are close to the original treatment effects (both the effect size and sign): one can only guarantee the similarity between observed outcomes but not counterfactuals.
- The technical contribution of this paper is not enough. The entire generation framework is based on the existing ADS-GAN model. The only difference is an additional contrastive loss to add penalty on the original loss function when the generated sample is much closer to x' than to x. The following potential outcome generation is based on a feed-forward neural network, which is a widely used backbone in causal effect prediction.
- The potential outcomes are generated directly through a neural network without any adjusting for confounding variables. Since the data is generated from real-world observational data that includes various confounding factors.
- Only three causal inference models are included in the experiments and two of them are from the same family (DR). More baselines (e.g., IPTW, matching-based methods) should be considered to examine the goodness of created dataset.
- Why propensity score stratification is much better than DR methods? When the results are not consistent with most studies, can we still trust the generated data as well as the ground truth?
- Only Wasserstein distance is used to measure the distance between original and synthetic dataset distribution. More metrics such as KL-divergence should be considered for comprehensive evaluation.
- What's the performance of three causal inference models on the original dataset? Are the results on the original dataset consistent with the results on the synthetic dataset?
- What's the influence of \beta in Eq.(3) to model performance? An ablation study should be conducted to show this influence.

**Summary Of The Paper:**

The paper studies the problem of generating synthetic patient data for the evaluation of causal inference models. The generated patient data is expected to highly mimic the distribution of the original dataset while also taking patient privacy into consideration. Experiments are conducted on the synthetic dataset with three well-established causal inference models.

**Summary Of The Review:**

An interesting paper, but it seems that the proposed problem is not well-addressed by the method.

---

> ### Author Response · Authors · 2021-11-20
> **Response to reviewer eqMK's comments**
>
> We’d like to thank the reviewer for spending time on our paper and providing feedback. Here is our response to the concerns.
>
> ### It is hard to guarantee that the generated treatment effects are close to the original treatment effects and counterfactuals.
>
> We agree with the reviewer that there is no such guarantee. And such guarantee in general will never exist, because the true treatment effects and counterfactuals cannot be known. To evaluate causal inference models, the perfect solution would be to use real patient covariates, true outcome response surface, and true causal effects to generate patient outcomes. This is not possible due to patient privacy concerns and the fact that true causal effects and outcome response surface cannot be known. The common practice in the literature is that researchers use arbitrary response functions and causal effects to generate patient outcomes for model evaluation purpose. Our work represents a significant improvement upon such practices: we generated pseudo-patients to preserve the complex joint distribution of patient variables and address the patient privacy concerns at the same time; we learned the response surface and the causal effects from realistic data and use them to generate patient outcomes. This is not the perfect solution, but we do think it is a significant improvement over practices just using arbitrary response surface and causal effects.
>
> ### The technical contribution of this paper is not enough.
>
> We agree our work does not significantly improve any machine learning models and algorithms. The novelty of our work is in the synthetic dataset we generated. We plan to release our synthetic dataset to researchers to evaluate their causal inference models studying the effects of anti-hypertensive drugs so that they have a large-scale dataset that (1) has known causal effects to compare their model results against, (2) mimics a large-scale patient cohort of more than 250, 000 hypertension patients’ multi-year history of diagnoses, medications, and laboratory values and (3) supports about 30 hypertension treatments. Although such a dataset is extremely fundamental and important, as far as we know, no such dataset (or anything similar to this) existed before our study.
>
> ### Only three causal inference models are included in the experiments and two of them are from the same family (DR). More baselines (e.g., IPTW, matching-based methods) should be considered to examine the goodness of created dataset.
>
> We think this is an excellent suggestion. We evaluated the IPTW and propensity matching model and reported the results in the updated manuscript.
>
> ### Why propensity score stratification is much better than DR methods? When the results are not consistent with most studies, can we still trust the generated data as well as the ground truth? Run the model on the original dataset ?
>
> This is an excellent point and prompted us to create a new hybrid dataset similar to the original dataset but with generated patient outcomes. We further evaluated the models with more performance metrics and more experiments. Please refer to the Results section in the updated manuscript for details.
>
> ### More metrics such as KL-divergence.
>
> We actually thought about this, but decided to only use Wasserstein distance. Calculating KL-divergence and Wasserstein distance for joint distributions in a high dimensional space is a non-trivial task. There are papers that study Wasserstein distance in such settings, like this one (which we follow to calculate Wasserstein distance in our paper),
>
> *Ishaan Gulrajani, Faruk Ahmed, Martin Arjovsky, Vincent Dumoulin, and Aaron C Courville. Improved training of wasserstein gans. In I. Guyon, U. V. Luxburg, S. Bengio, H. Wallach, R. Fergus, S. Vishwanathan, and R. Garnett (eds.), Advances in Neural Information Processing Systems, volume 30. Curran Associates, Inc., 2017*
>
> It is unclear to us how to calculate KL-divergence in a high dimensional space.

---

### Official Review · Reviewer_Jr99 · 2021-11-02

**Correctness:** 3
**Technical Novelty And Significance:** 2
**Empirical Novelty And Significance:** 2
**Recommendation:** 3
**Confidence:** 5

**Main Review:**

The paper is well-written and easy to read. It also addresses a very important, long-lasting problem in the literature; that is, generating datasets that include realistic counterfactuals. I really liked the idea of generating pseudo-patients to preserve privacy.
That being said, I think there is a fundamental problem with generating the counterfactuals from a model trained on data: this is the whole reason why we develop causal inference methods. If we already have an algorithm that can predict the true counterfactuals, then the problem has been solved (and we no longer need datasets for evaluating causal inference methods)! So we agree that we cannot be sure that the proposed algorithm predicts the true counterfactuals. Then, training causal inference methods using the generated dataset means they merely try to learn the underlying function that the proposed outcome neural network models. I’m not sure how this is different from adopting any arbitrary response surface as Hill et al. and ACIC organizers have done.
Please see my other comments below:
- Please provide a reference for the statement that the ACIC dataset is “limited by non-representative populations”. As far as I know, ACIC datasets are all adopted from large medical studies.
- Please define “identifiability” and “realisticity” formally.
- For calculating the weights, the authors have used the inverse of H. I think it’s wrong though: imagine a feature is the same for all patients, then the entropy for that feature would be zero, and the inverse of zero is infinity. So a feature that doesn’t matter at all (for identification) gets a large weight. Or am I missing something here?
- The paper provides little discussion on many design choices (e.g., why ADS-GAN or WGAN-GP was selected, etc.).
- I couldn’t understand why “the neural network weights for treatment connections can be interpreted as the causal treatment effects”. Also, the true individual treatment effects are often different from the average treatment effect (think personalized medicine)!

Minor comments:
- The abstract is too long; consider reducing it to one third.
- Wrong citation format; in most cases, both name and year must be inside parentheses.
- Citation for ADS-GAN is missing in the text.
- Page 5, line 2: don't → do not
- The “subset of” sign in page 5, lines 9, 10, and 15 should be reversed.


**Summary Of The Paper:**

This work proposes a method to generate a set of binary datasets that are realistic and include counterfactual outcomes so that they can be used for evaluating causal inference algorithms. Moreover, the proposed method ensures that the original patients’ privacy is preserved.


**Summary Of The Review:**

Neither of the two components of the paper (i.e., privacy preservation and counterfactual prediction) are novel (they are adopted from previous works). The claim that the proposed counterfactual prediction component generates true counterfactuals is not supported (and my understanding is that it cannot be supported either). For these reasons, I think the proposed algorithm is not practical and therefore, I recommend rejecting the paper.

---

> ### Author Response · Authors · 2021-11-20
> **Response to reviewer Jr99's comments**
>
> We appreciate the reviewer for spending time reading our paper and providing detailed feedback. Here is our response to the concerns.
>
> ### The claim that the proposed counterfactual prediction component generates true counterfactuals is not supported (and my understanding is that it cannot be supported either).
>
> We totally agree with the reviewer that true counterfactuals cannot be supported. We think we miscommunicated in the paper by using “True causal effect” in the title of Section 4.3 and in the last sentence of the first paragraph in Section 4.3 as well as some other places.
>
> To clarify, when we claim that we generated synthetic data with ground truth for the causal effects, the “ground truth” does not mean we captured the true causal effects, but means it is the grand truth in the context of evaluating causal inference models because the patient outcomes of the synthetic data are generated from these causal effects. Many researchers used arbitrary functions and arbitrary causal effects to generate such patient outcomes. We want to do better than that by learning from real data the mapping between covariates and the outcomes, as well as the causal effects, which might be biased estimates, but closer to the true effects than arbitrary values. We revised our manuscript and clarified this in the first paragraph of Section 4.3.
>
> ### Learning the underlying function that the outcome neural network models vs. learning an arbitrary function.
>
> This is a fair point. We argue that the response surface and the magnitude of the causal effects do affect model performance. For example, the paper below suggests the smaller the causal effects are (i.e., the closer the outcome surfaces are between the treated and control groups), the better their models perform.
>
> *T. Wendling et al, "Comparing methods for estimation of heterogeneous treatment effects using observational data from health care databases," Statistics in Medicine, vol. 37, no. 23, pp. 3309-3324, June 2018.*
>
> Therefore, although we cannot be sure the proposed neural network captures the real response surface and the true causal effects, we still want to make the response surface used in outcome generation as close to the true response surface as possible and use causal effects as close to the true causal effects as possible.
>
> ### Comments related to ACIC, definition of “identifiability” and “realisticity”, inverse of H, design choices.
>
> We agree with most of the comments and updated the manuscript accordingly: we updated our description of ACIC and provided references and defined “identifiability” and “realisticity” in the first paragraph of Section 4.2.  With regard to H vs the inverse of H, we believe the weight should be inverse of H. Please note that the this is the weight of the difference. In the reviewer’s example, if a feature is the same for all patients, any value that deviates from it, even by a very small amount, can be used to identify the patient (because it makes this patient stand out among all other patients), and so it should be given a large weight.  We elaborated the reason why we settled on ADS-GAN in the second paragraph of Section 4.2
>
> ### Why “the neural network weights for treatment connections can be interpreted as the causal treatment effects”
>
> Please note that in figure 1, the activation function for domain $X_{T}$ is linear.  For a treatment $t_{i}$, the outcome of the neural network can be expressed as $X_{0}(t_{i}/X_{C}) = g(X_{C}) + … + w_{i} * t_{i}$.  So the causal effect of $t_{i}$ is just $X_{0}(t_{i} = 1/X_{C}) – X_{0}(t_{i} = 0/X_{C}) = w_{i}$.  We can see that $w_{i}$ is indeed the causal effect of treatment $t_{i}$.
>
> The reviewer is right that true individual treatment effects are often different from the average treatment effect. In this paper the neural network does not aim to capture any interactions between treatments and patient variables, i.e, there are no causal effect modifiers. With this design, the ITE are the same as the ATE, simplifying the complexity of the study and making the scope manageable.
>
> We agree with all other comments and shortened the abstract, fixed citation format, cited the ADS-GAN paper properly, changed (don’t) to (do not), and corrected the “subset of” sign.

---

### Official Review · Reviewer_46uW · 2021-11-02

**Correctness:** 3
**Technical Novelty And Significance:** 2
**Empirical Novelty And Significance:** 2
**Recommendation:** 3
**Confidence:** 3

**Main Review:**

**Strengths**:
- The problem tackled is very relevant for causal inference problems, as there is almost always a lack of ground truth available to assess how well different causal inference methods perform.
- The paper is well-written and virtually error-free.

**Weaknesses**:
- When the inclusion and exclusion criteria are applied to the original dataset, the authors are potentially introducing selection bias by conditioning on downstream variables such as hypertension. This can then introduce spurious correlations between any common causes of the selected variables, which may for example include treatment variables if one of the treatments has an effect on hypertension. I was surprised to see that the authors have not at least discussed a potential selection bias issue. Furthermore, other data preprocessing steps such as filling in missing values or standardizing the values have the potential to make or break existing causal relationships.
- The experimental section is rather limited. The authors only consider a single starting dataset, and evaluate only a few methods for causal effect estimation on the synthetic data.

Other comments:
- It is possible I am missing something, but it seems to me that the generator and discriminator roles are switched. Based on the explanation before Definition 1, it appears as though the generator is maximizing the loss so as to ensure that the synthetic dataset is not too close to the original one in terms of $\epsilon$-identifiability, while the discriminator is minimizing the loss to ensure that $P_{\hat{X}}$ and $P_{X}$ are indistinguishable. Shouldn't it be the other way around?
- The ADS-GAN model and $\epsilon$-identifiability should have corresponding citations to Yoon et al. (2020) when first mentioned. Furthermore, it would be helpful to write what the acronym ADS-GAN stands for.
- I share the authors' concern that propensity score stratification performs better than the doubly robust methods. Is the propensity model the same for both methods? Perhaps there is some particular structure in the data that favors the simpler propensity score stratification.
- page 6, subsection 4.3, second paragraph: Treatment domain is introduced as $X_T$, yet is almost immediately replaced with $\mathcal{T}$? Are these two sets the same? If not, what is the difference? I cannot find where $\mathcal{T}$ is defined.
- page 6, last paragraph: New domain set $\mathbb{Y}$ for $Y$ is introduced out of the blue. Perhaps the authors meant $\mathbb{R}$ instead?
- page 8, third line after Figure 3: small typo, "We" should not be capitalized

**Summary Of The Paper:**

The authors propose a method for using real-world patient data to generate (semi-)synthetic privacy-preserving data on which to evaluate methods for causal effect estimation. For this purpose, they adapt the ADS-GAN (Anonymization through Data Synthesis using Generative Adversarial Networks) model introduced by Yoon et al. (2020) to produce a synthetic dataset that is identical in distribution to the original dataset, yet cannot be used to identify any of the original patients. To ensure the former desideratum, the authors calculate the Wasserstein distance between $P_{\hat{X}}$ and $P_X$; to ensure the latter, they use $\epsilon$-identifiabilty (Yoon et al., 2020). The authors show empirically that the synthetic dataset satisfies these desiderata, and then evaluate a few causal effect estimators on the synthetic data using the generated ground truth.

**Summary Of The Review:**

The authors have presented a simple and potentially interesting approach for generating privacy-preserving synthetic data from a real-world dataset, but I am concerned by the fact that selection bias was introduced when the authors applied inclusion and exclusion criteria on the patients from the original dataset. Furthermore, the method is only applied to one real-world dataset, so it remains unclear to me how well it can be applied to datasets with different characteristics (e.g. size, amount of missing data, data types).

---

> ### Author Response · Authors · 2021-11-20
> **Response to Reviewer 46uW's comments**
>
> We sincerely thank the reviewer for spending time reading our paper and providing detailed feedback. Below is our response to the concerns raised in the review.
>
> ### Selection bias induced by conditioning on downstream variables such as hypertension and data preprocessing
>
> We agree that both stratification on downstream hypertension and data preprocessing have the potential to induce bias in our datasets (both original and synthetic) relative to the true underlying data generating process. We updated the manuscript and discussed it in the Discussion section.
>
> We would like to further clarify three points with respect to this.
>
> * In this work, as our aim was to generate synthetic data to evaluate causal inference models that study causal effects of anti-hypertensive drugs, our target population is the patients affected by hypertension.
> * The original dataset we use as a template did include selection for individuals with hypertension. However, we only include time periods in the analysis subsequent to a hypertensive-qualifying events. This was designed with a target trial in mind in which patients are recruited at an initial qualifying measurement and then followed up after treatment assignment. We believe this minimizes the impact of selection bias due to conditioning on hypertension in our original data.
> * We do not claim that the causal effect estimates in the paper are unbiased estimates of the true causal effects. We claim that we generated synthetic data with ground truth for the causal effects, the “ground truth” does not mean we captured the true causal effects, but means it is the grand truth in the context of evaluating causal inference models because the patient outcomes of the synthetic data are generated from these causal effects. Many researchers used arbitrary functions and arbitrary causal effects to generate such patient outcomes. We want to do better than that by learning from real data the mapping between covariates and the outcomes, as well as the causal effects, which might be biased estimates, but closer to the true effects than arbitrary values. We further clarified this point in the first paragraph of Section 4.3 in the updated manuscript.
>
> ### Produced only a single dataset and evaluated only a few methods.
> This is a very good observation. We conducted additional experiments and evaluated two more causal inference models. Please see the updated manuscript for details. In the future, we plan to produce more datasets for multiple diseases and evaluate more methods. We do want to control the scope of this work by only focusing on hypertension. We chose hypertension because it affects almost half of adults in the United States and we think that a synthetic dataset that can be used to evaluate causal models studying anti-hypertensive drugs is of significant value by itself.
>
> ### Generator and discriminator are possibly switched.
> We believe our paper is right and accurate in this regard. Please refer to the paper below for detailed explanation.
>
> *Jinsung Yoon, Lydia N. Drumright, and Mihaela van der Schaar. Anonymization through data synthesis using generative adversarial networks (ADS-GAN). IEEE J. Biomed. Health Informatics, 24(8):2378–2388, 2020*
>
> ### Performance comparison between propensity score stratification and the doubly robust methods.
> The propensity model for propensity stratification is a logistic regression classifier. We agree this is an interesting observation. We conducted more experiments and evaluated more models to have a better picture of the relative performance of different models. Please refer to the Results section in the updated manuscript for detailed discussions.
>
>
> ### All other comments.
> To address all these concerns, we significantly shortened the abstract, cited the ADS-GAN paper in the first section, and corrected all the notation issues related to $\mathcal{T}$, $X_{C}$ and $\mathbb{Y}$

---

### Decision · Program_Chairs · 2022-01-20

**Decision:**

Reject

**Comment:**

In this paper, the authors propose a method for generating high quality synthetic datasets, and use their methods to evaluate a variety of causal effect estimators.

In general, the paper was not received very favorably by reviewers.  The primary concerns were: (a) issues with built-in bias in the algorithm that generates synthetic data (due to collider stratification bias induced by conditioning on causally "downstream" variables, (b) issues with "replicating underlying counterfactuals," which indeed is a difficult problem, and (c) lack of "technical novelty."

First, I am personally very sympathetic to what the authors are trying to do.  Regardless of current reviewer reception, I think the causal inference community really needs more high quality benchmarks, and (semi)synthetic datasets, and validation approaches.  I urge the authors to continue this line of work.

That said, I think it is important (for causal benchmarks) to be clear about the distinction between the observed data distribution (e.g. p(C,A,Y) for the backdoor model), and the full data distribution (e.g. p(C, A, Y(0), Y(1)) for the backdoor model with a binary treatment).

Generally what makes a benchmark interesting is preserving some features of the _full_ data distribution, and allowing "knobs" that make the problem easier and harder.  Much of what the ACIC competition organizers did was provide such knobs.  Mimicking features of just the observed data distribution, even if they are complicated, isn't enough to make a causal benchmark interesting, since the problem is all about how full and observed data relate.

When revising the paper, please keep this difference in mind, and consider what features of p(C, A, Y(0), Y(1)) (or more complex versions of this) make for an interesting benchmark, while also generating p(C,A,Y) that "mimics observed data" in some way.